# Protocol for a systematic review on routine use of antibiotics for infants less than 6 months of age with growth failure/faltering

Aamer Imdad ,[1] Fanny F Chen,[2] Melissa François,[2] Emily Tanner-Smith ,[3] Abigail Smith,[4] Olivia J Tsistinas,[4] Jai K Das ,[5] Zulfiqar Ahmed Bhutta[6,7]

For numbered affiliations see end of article.

**Correspondence to**
Dr Aamer Imdad;
imdada@upstate.edu

## ABSTRACT

**Introduction** Antibiotics have been used as an adjunct in treating children with severe acute malnutrition 6–59 months of age; however, the data for infants less than 6 months are scarce. The WHO recently started guideline development for preventing and treating wasting, including growth failure/faltering in infants less than 6 months. This systematic review commissioned by WHO aims to synthesise evidence from current literature on the effectiveness of antibiotics for infants less than 6 months of age with growth failure/faltering.

**Methods and analysis** We will conduct a systematic review and meta-analysis for studies that assessed the effect of antibiotics in the treatment of infants with growth faltering. We will search multiple electronic databases. We will include randomised control trials and non-randomised studies with a control arm. The study population is infants less than 6 months of age with growth failure. The intervention group will be infants who received no antibiotics or antibiotics other than recommended in 2013 guidelines by WHO to treat severe acute malnutrition in children. The comparison group will be infants who received antibiotics according to the 2013 guideline by WHO. We will consider the following outcomes: mortality, clinical deterioration, antimicrobial resistance, recovery from comorbidity, adverse events, markers of intestinal inflammation, markers of systemic inflammation, hospital-acquired infections, non-response. We will use the meta-analysis to pool the studies where applicable. We will use the Grading of Recommendations Assessment, Development, and Evaluation approach to reporting the overall evidence quality for an outcome.

**Ethics and dissemination** This is a systematic review and will not involve contact with a human subject. The findings of this review will be published in a peer-review journal and will guide the WHO's recommendation for the use of antibiotics in infants less than 6 months of age with growth failure.

**PROSPERO registration number** CRD42021277073.

## STRENGTHS AND LIMITATIONS OF THIS STUDY

⇒ We will search several electronic databases and include randomised and non-randomised studies.
⇒ Two authors will screen the titles and extract the data from included studies duplication. We will assess the risk of bias for each outcome from each study and use the Grading of Recommendation, Assessment, Development and Evaluation criteria to assess the overall quality of evidence.
⇒ We will conduct the meta-analysis if the data are available for more than one study for an outcome and there is clinical and methodological homogeneity in the included studies.
⇒ All the included studies may not report the data for all the outcomes mentioned in this review, which will be a limitation of the study.

## INTRODUCTION

The WHO and the United Nations Children's Fund (UNICEF) estimate that nearly 14 million children suffer from severe acute malnutrition (SAM) worldwide.[1] Infants less than 6 months of age are particularly vulnerable to the effects of inadequate nutrition. Higher mortality rates secondary to growth failure are seen in this age group compared with older infants and children.[2 3] Despite excess mortality risk and increasing prevalence of wasting in this population, limited studies exist to guide the management of young infants with growth failure and faltering.[2–6] Malnutrition not only increases the risk of severe infections in children and triples the risk of mortality from pneumonia, measles or diarrhoea but also detrimentally affects their neurodevelopment.[7–9] Therefore, the current practice for children 6 months to 5 years of age with SAM is to prescribe routine antibiotics, such as 5–7 days of amoxicillin, when they are inpatients.[9] However, current recommendations state that the same general medical care should be used for infants with SAM who are less than 6 months of age as infants above 6 months of age, even though there is limited evidence to support this recommendation for infants less than 6 months of age.[2]

Furthermore, even though antibiotics have been shown to be effective in children 6–59 months of age with SAM, this practice in infants has the potential to harm due to recently identified risks of antibiotic use in infancy, including the diminishment of infant gut microbiome,[10] future development of obesity, allergic disorders[11] and autoimmune disorders.[12] The urgency in more targeted management guidelines is further underscored by the physiological differences in renal and gastrointestinal function in infants compared with older children.[12] The WHO recently started the process of guideline development for the prevention and treatment of wasting in children, including growth failure/faltering in infants less than 6 months. We aim to systematically review and synthesise evidence from current literature on the effectiveness of antibiotics for infants less than 6 months of age with growth failure/faltering.

## REVIEW QUESTION

In infants <6 months with growth failure/faltering, what are the effects of no routine antibiotics or different approaches (eg, types of antibiotics, doses, etc) compared with routine antibiotics following treatment protocols in the 2013 WHO guidelines on the outcomes of interest?

## METHODS

This systematic review will be conducted according to methods described in Cochrane Handbook[13] and reported using Preferred Reporting Items for Systematic Reviews and Meta-Analyses guidelines 2020.[14] We registered the protocol on the PROSPERO website.

### Inclusion criteria

We will include randomised and non-randomised controlled trials (RCTs). We will consider both individual and cluster randomised trials. We will also consider non-randomised trials and cohort studies with a controlled arm. Table 1 gives the definitions of the eligible study designs. We will exclude case–control studies, case reports, case series and commentaries.

### Population

The population of interest is infants under 6 months of age with growth failure/faltering. As there is no standard definition of growth failure/faltering, we will consider author definitions. We will include studies irrespective if they are done in the community setting or the hospital setting. We will consider studies that included infants infected with HIV. We will include studies with infants born low birth weight or preterm; however, we will exclude studies on infants admitted to the neonatal intensive care units. We will exclude studies that included infants with congenital anomalies only. If a study includes infants less than 6 months and greater than 6 months of age, we will consider this study if the disaggregated data are available for infants less than 6 months. If the disaggregated data are not available from such a study, we will consider the study if the average age of infants is at or below 6 months of age.

### Intervention

We will include all antibiotic treatments given systemically, such as amoxicillin, augmentin, cephalosporins and macrolides. We will include studies irrespective of dosage, frequency, duration or route of administration; however, topical application of antibiotics will not be

| Table 1 | Suggested study designs for inclusion in the review | |
| --- | --- | --- |
| **Suggested terms** | **Definition** | **Notes** |
| Randomised controlled trial (RCT) OR randomised trial | An experimental study in which people are allocated to different interventions using methods that are random. | We will consider both individual and cluster randomised trials. We also consider the factorial design trials |
| Non-randomised controlled trial (NRCT) OR non-randomised trial | An experimental study in which people are allocated to different interventions using methods that are not random. | We will use consistent terminology and avoid using the term quasi-experimental as it might have different meanings in different settings. We will use the ROBINS-1 for risk of bias assessment for this study design |
| Controlled before-after study (CBA) | A study in which observations are made before and after the implementation of an intervention, both in a group that receives the intervention and in a control group that does not. | We will require two minimum criteria for the inclusion of CBAs.<br>► Data Collection: We will include CBAs if the data for the intervention and control groups were collected prospectively in the same time frame<br>► Choice of Control: We will include CBAs that include a control at a second site to avoid contamination of the intervention to the control group if the settings and populations are the same for the intervention and control groups. |

This table is modified from: https://epoc.cochrane.org/sites/epoc.cochrane.org/files/public/uploads/EPOCStudyDesignsAbout.pdf

considered. We will consider studies if the antibiotics were given empirically at the diagnosis of growth failure or faltering. We will also consider studies if the antibiotics were given in response to a suspected infection in infants with wasting. We will exclude studies where antibiotics were given for the prevention of wasting in otherwise healthy infants. We will also exclude studies where antibiotics were given for other reasons, such as suspected serious bacterial infections in otherwise healthy infants. We will allow cointervention such as nutritional supplementation for nutritional rehabilitation only if similar in both groups.

### Comparison
The comparison groups are routine antibiotics following treatment protocols detailed in the 2013 WHO guideline.[2]

### Outcomes
► Mortality (dichotomous outcome).
► Clinical deterioration (dichotomous outcome, defined by the development of any danger signs (obstructed breathing, respiratory distress, cyanosis, shock, severe anaemia, convulsion, severe dehydration, profuse watery diarrhoea, vomiting and/or impaired consciousness)).
► Antimicrobial resistance (dichotomous outcome, as defined by authors).
► Recovery from wasting (dichotomous outcome, as defined by authors).
► Recovery from comorbidity, pneumonia (dichotomous outcome).
► Recovery from comorbidity, diarrhoea (dichotomous outcome).
► Recovery from any comorbidity (dichotomous outcome).

► Adverse events—incidence of diarrhoea (dichotomous outcome, >3 loose stools/24 hours).
► Markers of intestinal inflammation—faecal calprotectin (continuous outcome).
► Markers of systemic inflammation—serum C reactive protein (continuous outcome).
► Hospital-acquired infections (dichotomous outcome).
► Non-response (eg, not achieving recovery within 4 months of initiating treatment) (dichotomous outcome).

We will consider all the outcomes at 1, 3 and 6 months and the longest follow-up. All the primary analyses will be done at the longest follow-up.

### Balance of benefits and harms
We will stratify the outcomes by benefit and harm and provide a narrative summary of the balance of benefits and harms.

### Literature search
We will conduct systematic electronic queries using key terms in multiple databases, including MEDLINE via PubMed, EMBASE, Web of Science, CINAHL, Scopus, LILACS, WHO Global Index Medicus and BIOSIS Previews. There will be no search restrictions on outcomes, publication year, publication status or publication language. The search strategies for different databases are available in online supplemental appendix 1. The references of formerly published reviews and recently published studies will be examined for potential inclusion. We will also use The Cochrane Central Register for Controlled Trials and ISRCTN registry to identify studies currently underway. We will also search the websites of pertinent international agencies such as the WHO (including WHO's Reproductive Health Library, electronic Library of Evidence of Nutrition Actions and Global database on the Implementation of Nutrition Action), UNICEF, Global Alliance for Improved Nutrition, International Food Policy Research Institute, International Initiative for Impact Evaluation, Nutrition International, World Bank, USAID and affiliates (eg, FANTA, SPRING) and the World Food Programme. We will search the abstracts of the major conference, such as annual paediatric academic society meetings. Finally, we will use the citation tracking function of the included studies in PubMed to look for any eligible studies.

### Data extraction and synthesis
Selection of studies and data extraction
　　We will use a three-phased approach to screen studies for eligibility. In the first phase, two authors will independently screen study titles and abstracts yielded from search results to identify potentially eligible ones. Studies selected during this initial phase will then go through a full-text review, which constitutes the second phase. In the final phase, studies that are determined to be eligible through the full-text review will undergo data extraction (figure 1). We will use the software Covidence in the screening process and create a form to store the

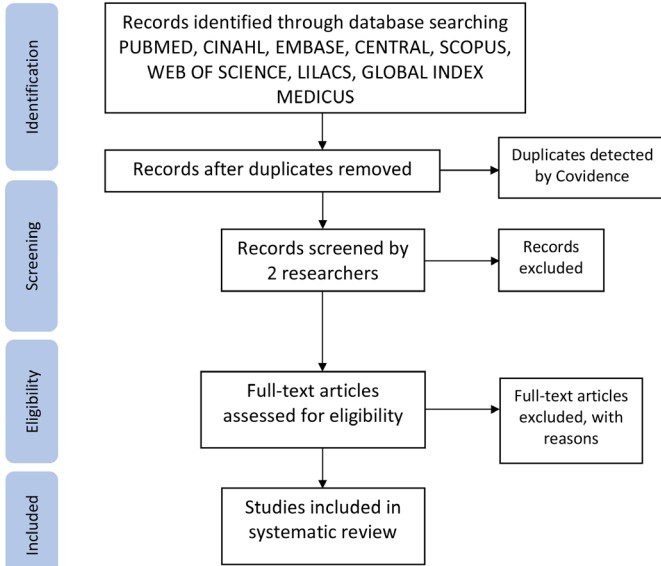

**Figure 1** PRISMA diagram of data extraction process. PRISMA, Preferred Reporting Items for Systematic Reviews and Meta-Analyses.

| Table 2 | Sample table to report the qualitative data from the included studies | | | | |
|---------|-----------------|----------------------|---------------------------|----------------------------|-------------------------|
| **Study** | **Study population** | **Intervention studied** | **Feasibility of intervention** | **Acceptability of intervention** | **Considerations for equity** |
| | | | | | |

data extracted in duplicate by two authors.[15] Two authors will screen the studies independently and compare their findings. The authors will resolve potential disagreements through discussions, and the senior author on the team may assist as needed.

We will extract characteristic data for each eligible study, including study region/country, study year, study type, intervention exposure (dose, duration, frequency), comparison, outcomes, population characteristics detailed in subgroup analysis and risk of bias. We will also extract the information on feasibility, acceptability, equity and resource use of the intervention and report this data in separate tables (tables 2 and 3). We will extract the raw values for the number of events in the intervention and control group in case of dichotomous outcomes. To avoid reviewer bias, we are deciding a priori the order of preference for extracting outcomes when data might be available in several formats. For randomised studies, we will prefer data that require the least manipulation by authors or inference by reviewers. We will give preference denominators in the following order for mortality data: number with the definite outcome known, number randomised and child-years. For morbidity outcomes to which both survivors and non-survivors may have contributed data (eg, the incidence of diarrhoea), we will give preference to child years/months/days, number with the definite outcome known and number randomised.

For continuous outcomes from randomised trials, we will extract the raw values (eg, means and SD) rather than calculated effect sizes (eg, Cohen's d) and calculate the summary estimate ourselves. For denominators, we will preference the numbers with the definite outcome known and then number randomised. For non-randomised studies, we will prefer to extract the summary estimates (mean difference or relative risk (RR)) adjusted for confounding variables.

If a study is only available in an abstract, we will contact the authors for full text. If the full text could not be obtained from any sources, we will include the abstract if sufficient details of study design and outcomes are available. We will try to find the protocol of each included study to assess the details of the methods. If the study protocol is not publicly available, we will contact the authors for the same. Suppose the results of the randomised trial are published in more than one report. In that case, we will

consider all the publications related to that study as one study but extract the data from all relevant publications.

### Assessment of risk of bias in included studies

We will evaluate the study risk of bias with the Cochrane risk of bias (ROB V.2.0) and Cochrane ROBINS-I for RCTs and non-randomised studies of interventions, respectively.[16 17] Two authors will assess the risk of bias separately and compare their findings and reach a consensus. In the case of conflicts, a discussion will be held to resolve the conflict. A senior author will be consulted if a consensus could be reached between the two primary authors.

The risk of bias assessment according to ROB-2 and ROBINS-I is done for each outcome and not for a particular study.[16 17] This risk of bias assessment is different from earlier practice where a study was assessed for risk of bias, and the same assessment was applied to all the outcomes extracted from that study.[13] We will summarise the risk of bias for each domain (such as sequence generation, allocation concealment, etc) and the overall risk of bias from the study for a particular outcome. The risk of bias will be summarised as high, low or some concerns. We will present the risk of bias assessment in the form of graphs.

### Data synthesis

We will report review findings both qualitatively and quantitatively. A narrative synthesis will be used to report the characteristics of all included studies and their results. A random effects meta-analysis will be employed when at least two studies possess sufficient clinical and methodological uniformity for synthesis. If quantitative synthesis is not appropriate, results will be described according to the standard guidelines of narrative synthesis.[18] The software RevMan will be used for statistical analysis.[19] Dichotomous outcomes will be assessed using RR effect sizes and presented with 95% CIs. The continuous outcomes will be combined to obtain a standardised mean difference and reported with a 95% CI. We will present the meta-analysis results in the form of forest plots and examine their statistical heterogeneity and inconsistency.

We will include the data from RCTs on an intention to treat analysis basis. If an RCT does not report an intention to treat analysis, we will generate our own intention to treat analysis by using the raw values extracted from the trial. Suppose an RCT does not report the raw values but

| Table 3 | Sample table for a description of qualitative data on resource use | | |
|---------|---------------------|-------------------------------|-----------------------------------------------|
| **Study** | **Intervention studies** | **Resources used for production** | **Resources used for administration of the intervention** |
| | | | |

reports the summary estimate, such as RR. In that case, we will include the summary estimate using the generic inverse variance weighting method of meta-analysis. For observational studies, we will prefer to use a summary estimate that was adjusted for confounding factors. If a non-randomised study reports an OR as the summary estimate, we will calculate the RR if the raw data are available. If the raw data are not available to calculate an RR, we will pool the OR if the large study sample size. We will do the sensitivity analysis by excluding such studies to see whether it has any detrimental effect on the final summary estimate of the pooled analysis.

We plan to pool the data separately from observational studies and randomised studies. Suppose the number of studies is small (less than 10). In that case, we may pool both the study designs in the same analysis but do a sensitivity analysis by excluding the observational studies.

We will consider the following two comparisons.
1. No routine antibiotics compared with routine antibiotics following treatment protocols in the 2013 WHO guidelines.
2. Different approaches to the use of antibiotics (eg, types of antibiotics, doses, etc) compared with the routine antibiotics following treatment protocols in the 2013 WHO guidelines.

### Studies with missing data
We will record missing data during the data extraction process. We will use imputations of missing data over recording the data as missing. If data are missing for some cases or dropouts are not reported, we will contact the study authors requesting the complete data. If a study does imputations for the missing data, we will use the imputed data. If a study does not report the raw values, we will

extract the summary estimates such as RR or risk ratio. If a study does not report mean but median, we will convert the median to mean based on the Cochrane Handbook of Systematic Reviews methods.[13] Suppose the SD is not available for a mean, and the value cannot be calculated from available data such as SE, CI or the p value. In that case, we will contact the authors for additional information. If the SD is not available from any source for a mean, we will use the SD from a similar study. If the data are presented only in graphs, we will extract the information to the best possible approximation.

### Assessment of heterogeneity
We will analyse statistical heterogeneity in the pooled data using $Tau,^2$ $\chi^2$ and $I^2$ statistics. We will also assess statistical heterogeneity through visual inspection of forest plots, using the $\chi^2$ test (assessing the p value) and calculating the $Tau^2$ and $I^2$ statistics. We will consider it significant statistical heterogeneity when the p value is less than 0.1, the $I^2$ value exceeds 50%, and the inspection of forest plots shows substantial variability in the effect of the intervention. Finally, we will perform subgroup analysis to identify reasons for eligible statistical heterogeneity.

### Assessment of reporting bias
We will assess publication bias of small studies using funnel plots and regression tests for funnel plot asymmetry when the meta-analysis includes at least 10 studies.

### Subgroup analysis and investigation of heterogeneity
► By different types/definitions of growth failure/faltering.
► Age at presentation (newborn (0–28 days), 1–3 months, 4–6 months).

| Table 4 | Description of methods for grading of overall evidence | | |
|---|---|---|---|
| **Study design** | **Quality of evidence*** | **Lower certainty score if** | **Higher certainty score if** |
| Randomised trial | High | Risk of bias | Large effect |
| | Moderate | ► 1 Serious | +1 Large |
| | | ► 2 Very serious | +2 Very large |
| Observational study | Low | Inconsistency | |
| | Very low | ► 1 Serious | |
| | | ► 2 Very serious | Dose–response |
| | | Indirectness | +1 evidence of a gradient |
| | | ► 1 Serious | |
| | | ► 2 Very serious | |
| | | Imprecision | All plausible confounding would: |
| | | ► 1 Serious | +1 Reduce a demonstrated |
| | | ► 2 Very serious | effect |
| | | Publication bias | |
| | | ► 1 Likely | |
| | | ► 2 Very likely | +1 Suggest a spurious effect when results show no effect |

*In the GRADE approach, randomised controlled trials (RCTs) start as high-quality evidence and observational studies as low-quality evidence supporting estimates of intervention effects. Five factors may lead to rating down the quality of evidence, and three factors may lead to rating up. Ultimately, the quality of evidence for each outcome falls into one of four categories, from high to very low. GRADE is 'outcome centric': rating is made for each outcome, and quality may differ indeed, is likely to differ from one outcome to another within a single study and across a body of evidence.[21]

- Gestational age: preterm birth (<37 weeks) versus full-term birth (>37 weeks).
- Birth weight: low birth weight (<2500 g) versus normal birth weight (>2500 g).
- HIV exposure: studies with participants exposed to HIV versus studies with no HIV exposure.
- Presentation: participants with oedema versus patricians with no oedema.
- Comorbidities: with or without comorbidities
- Nutrition: babies breast feeding or non-breastfed babies
- Location of the treatment: inpatient or outpatient/community
- Dose of antibiotics
- Duration of antibiotics: 7 days versus >7 days
- Type of antibiotics

The subgroup analyses above will be based on individual-level data. We will try to extract the information from individual studies as available. If the information is not available from the published studies, we will write to the authors to get the required data.

### Sensitivity analysis

We will complete sensitivity analysis by removing studies with a high risk of bias. We will compare results from random versus fixed-effect meta-analysis models.

### Rating of overall quality of evidence

The Grading of Recommendations Assessment, Development and Evaluation (GRADE) approach will be used to evaluate the overall certainty of evidence using the software GRADEpro.[20] The GRADE approach is a comprehensive framework used to assess the overall certainty of the evidence for an outcome using study characteristics such as study design, inconsistency, indirectness of evidence, risk of bias, publication bias and imprecision estimates. We will include the GRADE assessment results in a GRADE evidence profile that contains certainty ratings, including very low, low, moderate or high based on the evidence across studies for primary outcomes (table 4).

### Amendments

We will conduct literature searches, screening of titles, selection of studies, data extraction and analysis according to the plan aforementioned in the protocol. We will report any additional analysis performed in the Methods section if such occurs.

**Author affiliations**
[1]Division of Pediatric Gastroenterology, Department of Pediatrics, SUNY Upstate Medical University, Syracuse, New York, USA
[2]College of Medicine, SUNY Upstate Medical University, Syracuse, New York, USA
[3]College of Education, University of Oregon, Eugene, Oregon, USA
[4]Health Science Library, SUNY Upstate Medical University, Syracuse, NY, USA
[5]Division of Women and Child Health, Aga Khan University, Karachi, Pakistan
[6]Division of Women and Child Health, The Aga Khan University, Karachi, Sindh, Pakistan
[7]SickKids, Toronto, Ontario, Canada

**Acknowledgements** We are very thankful to Allison Daniel, Jaden BENDABENDA, Kirrily De POLNAY and Zita WEISE PRINZO for their input to improve this protocol.

**Contributors** AI and ZAB conceptualised the study. FFC and AI wrote the first draft of the manuscript. MF contributed to method. ET-S contributed to methods. AS and OT designed the search strategy. JD, ZAB provided the supervision as experts and contributed to methods and manuscript writing.

**Funding** This work is funded by World Health Organization grant number 202725572. WHO also provided technical support for this work.

**Competing interests** None declared.

**Patient and public involvement** Patients and/or the public were not involved in the design, or conduct, or reporting, or dissemination plans of this research.

**Patient consent for publication** Not applicable.

**Provenance and peer review** Not commissioned; externally peer reviewed.

**ORCID iDs**
Aamer Imdad http://orcid.org/0000-0002-7026-0006
Emily Tanner-Smith http://orcid.org/0000-0002-5313-0664
Jai K Das http://orcid.org/0000-0002-2966-7162

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
