## [Reviewer comments · BMJ Open]

ARTICLE DETAILS

TITLE (PROVISIONAL)	A Protocol for a Systematic Review on Routine Use of Antibiotics for Infants Less Than Six Months of Age with Growth Failure/Faltering
AUTHORS	Imdad, Aamer; Chen, Fanny F; François, Melissa; Tanner-Smith, Emily; Smith, Abigail; Tsistinas, Olivia; Das, Jai K.; Bhutta, Zulfiqar

VERSION 1 – REVIEW

REVIEWER	Francisco Reyes-Santías Universidad de Vigo, Organización de Empresas e Mercadotecnia
REVIEW RETURNED	07-Jan-2022

GENERAL COMMENTS	REVIEW REPORT FOR THE STUDY "ROUTINE ANTIBIOTICS FOR INFANTS LESS THAN SIX MONTHS OF AGE WITH GROWTH FAILURE/FALTERING. A PROTOCOL FOR THE SYSTEMATIC REVIEW" Journal: BMJ Open Along with the paper "Routine antibiotics for infants less than six months of age with growth failure/faltering. A protocol for the systematic review", the authors conducted a systematic review and meta-analysis of studies investigating the efficacy of antibiotics in the treatment of infants with growth disorders. Randomized control trials and non-randomized trials with a control arm are included. The study population consists of infants less than six months of age with growth disorders. Intervention group. The intervention group has consisted of infants who had received no antibiotics or antibiotics other than those recommended in 2013. The control group has consisted of infants who received antibiotics according to the 2013 WHO guidelines for the treatment of severe acute malnutrition in children. Title and summary. The title and abstract express well the object of study, objectives, and results of the article. Structure of the article: The contents are well organized and they adhere to the IMRaD structure. It includes a theoretical framework of the research problem but at this point, I suggest the authors incorporate some other bibliographic references that I miss in the text: Brander RL, Pavlinac PB, Walson JL, et al. Determinants of linear growth faltering among children with moderate-to-severe diarrhea in the Global Enteric Multicenter Study. BMC Med. 2019;17(1):214. Published 2019 Nov 25. doi:10.1186/s12916-019-1441-3 Shekhar S, Petersen FC. The Dark Side of Antibiotics: Adverse Effects on the Infant Immune Defense Against Infection. Front Pediatr. 2020;8:544460. Published 2020 Oct 15. doi:10.3389/fped.2020.544460 Focusing on the opportunity of the study, it must be said that it is useful work since it covers one of the major problems resulting from the treatment of infants with growth disorders.
--

	Materials and methods. Regarding the material and methods section, the methodology is tailored to the object of study and the objectives and is explained in a transparent manner while it has been validly applied to guarantee the results. Results. The results are significant and they are presented in an adequate and understandable way, however, I would suggest incorporating a flowchart of the paper selection process as well as a forest plot. The results justify and relate to the objectives and methods and the results are of sufficient interest. Discussion. The discussion appropriately compares the study results with other works, highlighting the main study findings. However, I would propose the inclusion of two bibliographic references in the discussion section: Nasrin D, Blackwelder WC, Sommerfelt H, et al. Pathogens Associated With Linear Growth Faltering in Children With Diarrhea and Impact of Antibiotic Treatment: The Global Enteric Multicenter Study. J Infect Dis. 2021;224(12 Suppl 2):S848-S855. doi:10.1093/infdis/jiab434 Prado EL, Larson LM, Cox K, Bettencourt K, Kubes JN, Shankar AH. Do effects of early life interventions on linear growth correspond to effects on neurobehavioural development? A systematic review and meta-analysis. Lancet Glob Health. 2019;7(10):e1398-e1413. doi:10.1016/S2214-109X(19)30361-4 Bibliography. The 47.05% of the bibliography cited in the study belongs to the previous five years. Overall, it is an interesting study and should be considered for publication in Int. J. BMJ Open, once the minor revisions proposed have been resolved.
--	--

REVIEWER	Trond Flaegstad University Hospital of North Norway, Pediatrics and Neonatal Medicine
REVIEW RETURNED	26-Apr-2022

GENERAL COMMENTS	Very good manuscript
----------------------

VERSION 1 – AUTHOR RESPONSE

Reviewer: 1

Dr. Francisco Reyes-Santías, Universidad de Vigo, Servicio Galego de Saude

Comments to the Author:

REVIEW REPORT FOR THE STUDY "ROUTINE ANTIBIOTICS FOR INFANTS LESS THAN SIX MONTHS OF AGE WITH GROWTH FAILURE/FALTERING. A PROTOCOL FOR THE SYSTEMATIC REVIEW"

Journal: BMJ Open

Along with the paper "Routine antibiotics for infants less than six months of age with growth failure/faltering. A protocol for the systematic review", the authors conducted a systematic review and meta-analysis of studies investigating the efficacy of antibiotics in the treatment of infants with growth disorders. Randomized control trials and non-randomized trials with a control arm are included. The study population consists of infants less than six months of age with growth disorders. Intervention group. The intervention group has consisted of infants who had received no antibiotics or antibiotics

other than those recommended in 2013. The control group has consisted of infants who received antibiotics according to the 2013 WHO guidelines for the treatment of severe acute malnutrition in children. Title and summary. The title and abstract express well the object of study, objectives, and results of the article.

Authors comment: Thank you for the comment. We appreciate your time to review our work so comprehensively

Structure of the article: The contents are well organized and they adhere to the IMRaD structure. It includes a theoretical framework of the research problem but at this point, I suggest the authors incorporate some other bibliographic references that I miss in the text:

Brander RL, Pavlinac PB, Walson JL, et al. Determinants of linear growth faltering among children with moderate-to-severe diarrhea in the Global Enteric Multicenter Study. *BMC Med.* 2019;17(1):214. Published 2019 Nov 25. doi:10.1186/s12916-019-1441-3

Shekhar S, Petersen FC. The Dark Side of Antibiotics: Adverse Effects on the Infant Immune Defense Against Infection. *Front Pediatr.* 2020;8:544460. Published 2020 Oct 15. doi:10.3389/fped.2020.544460

Authors comment: Thank you for the suggestion. We reviewed and added the recommended articles where appropriate.

Focusing on the opportunity of the study, it must be said that it is useful work since it covers one of the major problems resulting from the treatment of infants with growth disorders.

Authors comment: Thank you for the comment. We can't agree more and we are excited to further this research.

Materials and methods.

Regarding the material and methods section, the methodology is tailored to the object of study and the objectives and is explained in a transparent manner while it has been validly applied to guarantee the results.

Authors comment: Thank you for the comment. We make every attempt to present clear and easy to follow protocols for our peers to review and study.

Results.

The results are significant and they are presented in an adequate and understandable way, however, I would suggest incorporating a flowchart of the paper selection process as well as a forest plot.

Authors comment: Thank you for the suggestion. We included a PRISMA flowchart for our paper selection process.

The results justify and relate to the objectives and methods and the results are of sufficient interest.

Authors comment: Thank you for the comment.

Discussion.

The discussion appropriately compares the study results with other works, highlighting the main study findings. However, I would propose the inclusion of two bibliographic references in the discussion section:

Nasrin D, Blackwelder WC, Sommerfelt H, et al. Pathogens Associated With Linear Growth Faltering

in Children With Diarrhea and Impact of Antibiotic Treatment: The Global Enteric Multicenter Study. J Infect Dis. 2021;224(12 Suppl 2):S848-S855. doi:10.1093/infdis/jiab434

Prado EL, Larson LM, Cox K, Bettencourt K, Kubes JN, Shankar AH. Do effects of early life interventions on linear growth correspond to effects on neurobehavioural development? A systematic review and meta-analysis. Lancet Glob Health. 2019;7(10):e1398-e1413. doi:10.1016/S2214-109X(19)30361-4

Authors comment: Thank you for the suggestion. We have reviewed and included these studies where appropriate in the introduction section as we do not have a discussion section due to the nature of our protocol.

Bibliography.

The 47.05% of the bibliography cited in the study belongs to the previous five years.

Overall, it is an interesting study and should be considered for publication in Int. J. BMJ Open, once the minor revisions proposed have been resolved.

Authors comment: Thank you for taking the time to carefully review our manuscript, we appreciate your feedback.

Reviewer: 2

Dr. Trond Flaegstad, University Hospital of North Norway

Comments to the Author:

Very good manuscript

Authors comment: Thank you for the review.

VERSION 2 – REVIEW

REVIEWER	Francisco Reyes-Santías Universidad de Vigo, Organización de Empresas e Mercadotecnia
REVIEW RETURNED	08-Jun-2022
GENERAL COMMENTS	Dear Editor, After seeing the authors' review my advice to the editors of BMJ Open is to publish the article "A Protocol for a Systematic Review on Routine Use of Antibiotics for Infants Less Than Six Months of Age with Growth Failure/Faltering". Yours sincerely, Francisco Reyes-Santias